# Production of Fungal Quinones: Problems and Prospects

**DOI:** 10.3390/biom12081041

**Published:** 2022-07-28

**Authors:** Johan Vormsborg Christiansen, Thomas Ostenfeld Larsen, Jens Christian Frisvad

**Affiliations:** Department of Biotechnology and Biomedicine, Technical University of Denmark, 2800 Kongens Lyngby, Denmark; jovoc@dtu.dk (J.V.C.); tol@bio.dtu.dk (T.O.L.)

**Keywords:** HPLC, mass spectrometry, quinone, toluquinone, terreic acid, anthraquinone, anthraquinone dimer, *Penicillium*, *Aspergillus*, *Talaromyces*

## Abstract

Fungal quinones can be used for a variety of applications, such as pharmaceuticals, food colorants, textile dyes, and battery electrolytes. However, when producing quinones by fungal cultivation, many considerations arise regarding the feasibility of a production system, such as the quinone yield, purity, ease of extraction, and the co-production of mycotoxins. In this work, we display the initial screening of filamentous fungi for quinone production and evaluate their potential for future optimization. We investigated toluquinone (TQ) potentially produced by *Penicillium* cf. *griseofulvum*, terreic acid (TA) produced by *Aspergillus parvulus* and *A. christenseniae*, and anthraquinone (AQ) monomers and dimers produced by *Talaromyces islandicus*. The strains grew on various agar and/or liquid media and were analyzed by ultra-high-performance liquid chromatography–diode array detection–quadrupole time-of-flight mass spectrometry (UHPLC-DAD-QTOF MS). In the case of AQs, feature-based molecular networking (FBMN) was used for the identification of AQ analogs. TQ was not observed in the production strains. TA constituted one of the major chromatogram peaks and was secreted into the growth medium by *A. parvulus*. The AQs constituted many major chromatogram peaks in the mycelium extracts and endocrocin and citreorosein were observed extracellularly in small amounts.

## 1. Introduction

Fungi are well known for their production of a wide range of secondary metabolite quinones, which are used as antibiotics [1,2,3], for sun protection [4], for degradation of plant matter [5], and more [6]. Most fungal quinones are synthesized by non-reducing and partially reducing polyketide synthases and often undergo several oxidation steps, resulting in compounds rich in oxygen [7,8]. They are able to undergo reversible two-electron transfer reactions, resulting in three charge-stages: the quinone stage (Q), a semi-quinone radical stage (QH^•^), and a fully reduced hydroquinone/quinol stage (QH_2_) [9,10]. While some quinones, such as phoenicin [11], are secreted out of the mycelium into the surroundings, other quinones are known to accumulate in fungal structures, such as the fusarubins, which accumulate in the perithecia of some fungi [12].

Much research has been conducted on the prospects of using quinones for various industrial purposes, such as pest control agents [13], dyes and colorants [14], pharmaceuticals [15], and as electrolytes in batteries [16]. Additionally, many filamentous fungi are good cell factories for the production of various metabolites [17]. However, even though many fungal quinones show high potential for various applications, little work has focused on the production aspects, with a few exceptions such as the patented Arpink Red™ (Natural Red™) [18] and microbial pigments, wherein quinones are sometimes considered [19].

In our previous work, we studied the production of the fungal quinone phoenicin and found that it could be produced in g/L levels with a relatively high degree of purity by wild-type *Penicillium* species [11]. Thus, we believe that it is worthwhile to investigate the production of other fungal quinones for potential high titers. However, when producing fungal quinones, many challenges can be encountered such as low yield and purity, mycelium extraction steps, and the potential co-production of mycotoxins.

In the work presented in this manuscript, we evaluate the prospects and challenges regarding the production of fungal quinones, using three types of quinones as models. In the first part of the paper, the production of toluquinone (TQ, Figure 1, **1**) is highlighted, a common shunt product of other secondary metabolite pathways originating from 6-methylsalisylic acid (6-MSA) [8,20]. Next, we evaluate the production of terreic acid (TA, **2**), a quinone epoxide produced by *Aspergillus* species from the sections *Terrei* and *Cervini* [6,21]. In the last part of the work, we showcase the production of anthraquinones (AQs, **3**), a well-studied quinone class with many industrial uses [22,23,24,25]. Each of these quinone examples highlights a different outcome when working with fungal production of metabolites, and the proposed next step for a better production is different for each case.

## 2. Materials and Methods

### 2.1. Fungal Strains

All fungal strains used in this work were from the IBT culture collection at the Technical University of Denmark (IBT Culture Collection of Fungi-DTU Bioengineering). The IBT numbers of each of the strains are shown in Table 1. Strains of different species and genera are included and the quinones produced by the fungi are very much related to the taxonomy. The *P*. cf. *griseofulvum* strains were chosen for the production of TQ, as preliminary laboratory experiments determined their ability to produce patulin, thus hypothesizing that the shunt products from that biosynthetic pathway (e.g., TQ) might also be present. The *T. islandicus* strains were selected, as this species is known for producing a plethora of both AQ monomers and dimers [6]. The *Aspergillus* strains selected were chosen as both are known to produce terremutin [26], one of the precursors to TA [27].

### 2.2. Fungal Cultivation and Extraction

#### 2.2.1. Growth Media

The growth media Czapek yeast autolysate agar (CYA), malt extract autolysate agar (MEA), malt extract autolysate agar oxoid brand (MEA-OX), potato dextrose agar (PDA), and yeast extract sucrose agar (YES) were prepared according to [28]. Liquid culture was made using the same procedures without the addition of agar.

#### 2.2.2. Cultivation on Agar Plates

When grown on agar plates, fungi were 3-point inoculated from a spore suspension and incubated for 7 days in the dark at 25 °C, except the *T. islandicus* strains, which were incubated for 10 days instead. Small-scale extraction of metabolites was carried out with a modified version of the plug extraction method, described by Smedsgaard [29]: One of the three colonies on an agar plate was extracted and no biological duplicates were made, as quantification was not a goal of this study. For most of the strains, 6 plugs were cut from one of the colonies on an agar plate with a 6-mm-diameter plastic straw and transferred to an Eppendorf tube. However, for the two *T. islandicus* strains, two types of extracts were made: a “colony extract” and an “agar extract”. For the colony extract, 3 plugs were cut from the mycelium colony, as described by Smedsgaard [29]. For the agar extract, three plugs were cut from the agar directly adjacent to the mycelium colony but without any fungal material. The following extraction steps were the same regardless of the plug extraction parameters. 1 mL extraction solvent (isopropanol:ethylacetate 1:3 *v*/*v* with 1% formic acid) was added and the plugs were ultrasonicated for 1 h. The solvent was transferred to a new Eppendorf tube and evaporated to dryness under a stream of nitrogen. The dried extract was re-dissolved in 100 μL methanol and ultrasonicated for 10 min before being centrifuged. The centrifugation steps always followed the following protocol: 3 min on a VWR MicroStar12 tabletop centrifuge at 11,400 × *g*. In total, 80 μL was transferred to a vial and were analyzed by ultra-high-performance liquid chromatography hyphenated to a diode array detector and a quadrupole time-of-flight mass spectrometer (UHPLC-DAD-QTOF).

#### 2.2.3. Cultivation on Liquid Media

Two of the strains, *A. parvulus* and *A. christenseniae*, were inoculated on liquid media and were grown as surface cultures. Each strain was grown in duplicates. A spore suspension of both strains was made by 3-point inoculating the strains on a PDA agar plate and incubating them for 7 days. After incubation, approximately 1 mL spore suspension (0.5 g tween 80, 0.5 g agar, 1 L water, autoclaved) was added and spores were scraped loose using a Drigalski spatula and transferred to a sterile glass vial. Spores were added with an inoculation needle to 15 mL medium in 50-mL sterile Falcon tubes with filtered lids to allow oxygen transfer. The Falcon tubes were tilted approximately 45 degrees to increase the surface area of the liquid, and fungi grew at 25 °C in the darkness for 11 days.

After incubation, supernatant and mycelium were extracted separately. Supernatant was isolated by decanting into a new Falcon tube. In total, 0.5 mL supernatant was transferred to a new Eppendorf tube and 0.5 mL acetonitrile was added. The samples were centrifuged and transferred to a new vial and analyzed by UHPLC-DAD-QTOF. These samples were not very concentrated and further extraction steps on the supernatant were performed. However, when performing tandem mass spectrometry analysis (MS/MS) on the TA analytical standard, one of the un-concentrated samples was used as a reference.

The continued extraction steps of the supernatant were carried out as described: pH of the supernatant was adjusted to 2 with hydrochloric acid and 15 mL ethyl acetate was added to the falcon tubes. After a few minutes, the ethyl acetate phase was transferred to a new Falcon tube and another 15 mL ethyl acetate was added to the supernatant sample. After a few minutes, the two organic phases were combined and evaporated under a flow of nitrogen until dryness. The dried extract was re-dissolved in 1 mL methanol, centrifuged, and 200 μL was transferred to a new vial and subsequently analyzed by UHPLC-DAD-QTOF.

The mycelium was extracted by adding 15 mL extraction solvent and ultrasonicated for 1 h. The solvent was transferred to a new Falcon tube and evaporated to dryness under a flow of nitrogen before being re-dissolved in 1 mL methanol. The extract was centrifuged and 200 μL was transferred to a new vial and was analyzed on UHPLC-DAD-QTOF.

### 2.3. UHPLC-DAD-QTOF-MS

UHPLC-DAD-QTOF analysis was carried out on an Agilent Infinity 1290 UHPLC system, hyphenated to a DAD and an Agilent 6545 QTOF mass spectrometer. Ionization was carried out with an Agilent Dual Jet Stream electrospray operated in either positive (+ESI) or negative mode (-ESI). MS/MS spectral data were acquired at three different collision-induced dissociation (CID) energies: 10, 20, and 40 eV. The method used has been described previously [30].

### 2.4. Data Pre-Processing

UHPLC-DAD-QTOF data was converted from Agilent “.d” formatting to “.mzML” with MSConvert (version 3.0, ProteoWizard, Palo Alto, California, USA) [31] and data peak picking was achieved with MZMine (version 2.53) [32]. The MZMine workflow was as follows: Mass detection was carried out at the MS1 level and MS2 level with noise level thresholds of 10^3^ and 10^2^, respectively. Chromatograms were made with the ADAP Chromatogram Builder Module [33] with the following parameters: Min group size in # of scans was set to 5, group intensity threshold and Min highest intensity were both set to 1000, and *m*/*z* tolerance was set to 10 ppm. The chromatogram deconvolution module was used with the local minimum search algorithm with a chromatographic threshold of 5%, a search minimum in the RT range at 0.1 min, a minimum relative height of 5%, a minimum absolute height of 5000, a Min ratio of peak top/edge at 5, and a peak duration range of 0.01 to 5.00 min. The *m*/*z* center calculation was set to MEDIAN. The Isotopics peaks grouper module was used with an *m*/*z* tolerance of 10 ppm, retention time tolerance of 0.1 min, the monotonic shape function set to false, a maximum charge of 2 and the representative isotope set to the most intense. Alignment was achieved with the Join aligner function with an *m*/*z* tolerance of 10 ppm, a weight for *m*/*z* at 20, a retention time tolerance of 5%, and a weight for RT at 20. The Require same charge state, Require same ID, and the Compare spectra similarity functions were set to false and the Compare isotope pattern function were set to True. The aligned feature list was exported by the Export/Submit to “GNPS-FBMN” module with the Merge MS/MS (experimental) function ticked with the following parameters: Select spectra to merge was set to across samples, the *m*/*z* merge mode was set to weighted average (remove outliers), the intensity merge mode was set to sum intensities, the expected mass deviation was set to 10 ppm, the cosine threshold was set to 70%, the peak count threshold was set to 20%, the isolation window offset (*m*/*z*) was set to 0, and the isolation window width (*m*/*z*) was set to 3.

### 2.5. Molecular Networking and Spectral Library Search

Feature-based molecular networking (FBMN) [34] was generated with the pre-processed data on the Global Natural Products Social Molecular Networking (GNPS) website (GNPS. Available online: https://gnps.ucsd.edu (accessed on 13 July 2022) [35]. First, MS/MS fragment ions were filtered if their mass was within 17 Da of the precursor *m*/*z*. A second filtering step of the MS/MS spectra was applied, choosing the top six fragment ions within a +/− 50 Da window. The ion tolerance of the precursor mass and the MS/MS fragment ions was set to 0.02 Da. The network edges were set to have a cosine score of 0.6 or above, with a minimum of six matched peaks. If two connected nodes appeared in the top 10 most similar nodes of each other, the edge between them was kept. The size of each molecular family was set to a maximum of 100. The network was used to search against the GNPS spectral library [35,36], with a score threshold of 0.7 and a minimum of five matching peaks. Annotation of the MS/MS spectra was carried out with the DEREPLICATOR tool [37]. Visualization of the network was carried out with Cytoscape [38].

Metabolite dereplication was also carried out with an in-house MS/MS spectral library, comparing experimental spectra with the spectra from the analytical standards. The comparison for each metabolite was carried out across three CID energies: 10, 20, and 40 eV. Feature picking before the analysis was carried out with the Agilent MassHunter Qualitative Analysis software (version B.07.00) using the function “Find by Auto MS/MS”. The retention time window was set to 0.25 min and the MS/MS total ion chromatogram threshold was set to 1000 counts with the mass match tolerance set to 0.05 Da. The forward score and reverse score of library matching was set to 25 and 80, respectively, and an accurate mass tolerance of 10 ppm for precursor ions and 20 ppm for fragment ions was used.

## 3. Results

### 3.1. Toluquinone

TQ is a small benzoquinone and, together with its hydroquinone form, toluquinol (4), is a shunt product of the patulin (5) mycotoxin pathway [8,20,39] (Figure 2). We investigated three patulin-producing *Penicillium* cf. *griseofulvulm* isolates for their ability to produce TQ and toluquinol alongside patulin. The strains were grown on CYA, MEA, MEA-OX, PDA, and YES before organic extraction followed by UHPLC-DAD-QTOF analysis.

Patulin was present in all three strains across all media as identified by MS/MS comparison to an in-house metabolite MS/MS spectral library across three CID energies: 10, 20, and 40 eV (Appendix A).

Patulin ionized poorly in +ESI but was one of the major peaks in the DAD chromatograms (Figure 3), showing that the strains were competent patulin producers. Next, we investigated the presence of TQ and toluquinol by searching for masses corresponding to the [M+H]^+^, [M+Na]^+^, [M+K]^+^, and [M+NH_4_]^+^ of those metabolites. In most samples, peaks corresponding to TQ and toluquinol were observed; however, these peaks were eluting prior to patulin, which is unexpected as patulin is more polar (reversed-phase chromatography was used). Thus, neither TQ nor toluquinol were found when the strains were grown in any of the studied growth media, showing the difficulty in the detection of biosynthetic shunt products. Conclusively, none of the tested strains were good producers of TQ according to the detection methods used.

### 3.2. Terreic Acid

#### 3.2.1. Cultivation on Agar Growth Medium

Two strains were investigated for the production of TA from *Aspergillus* section *Cervini*, *A. parvulus*, and *A. christenseniae* [40], a section that is known for its production of terreic acid [6]. Both strains grew on MEA and MEA-OX and extracts were analyzed by +ESI, -ESI, and DAD. TA was observed in the extracts from both strains on all the tested media when analyzed in –ESI but were absent in +ESI (Figure 4A,B). The corresponding TA peak with DAD detection constituted one of the major peaks (Figure 4C). One of the largest peaks across all detection methods had an accurate mass of *m*/*z* 157.0497 and *m*/*z* 155.0358 for +ESI and –ESI, respectively, corresponding to the [M+H]^+^ and [M-H]^−^ adducts of a compound with the molecular formula C_7_O_4_H_8_ (156.0424 Da). This formula corresponded to terremutin (Figure 4D, **6**), one of the precursors to TA [39]; however, the identity could not be verified as no data was available in our in-house MS/MS spectral library on terremutin. TA was identified by MS/MS spectral comparison to a TA analytical standard (Appendix A).

#### 3.2.2. Cultivation in Liquid Medium

The production of TA by *A. parvulus* and *A. christenseniae* was investigated on liquid growth medium to evaluate whether the quinone was secreted into the growth medium or whether it was deposited intracellularly or in the cell wall. Both strains were inoculated in duplicates; however, only one *A. christenseniae* duplicate grew. The two *A. parvulus* duplicates and the one *A. christenseniae* duplicate that grew formed a thick mycelium mat on top of the medium. The chromatograms of the two *A. parvulus* replicates looked very similar for both the supernatant and the mycelium extracts after UHPLC-DAD-QTOF analysis, thus only one of these was further analyzed.

For *A. parvulus*, TA was only observed in the supernatant extract while the putative terremutin was present in both the mycelium and supernatant extracts (Figure 5A,B). In contrast, neither metabolites were observed in the mycelium or supernatant extracts of *A. christenseniae* (Figure 5C,D). TA was one of the major compounds present in the supernatant extract of *A. parvulus* according to the DAD chromatogram (Figure 5E). This shows one of the advantages of testing more than one production species.

### 3.3. Anthraquinones

*T. islandicus* is well known for its production of AQs, including AQ dimers [6]. Thus, we investigated two strains from this species for their potential to produce AQs. The two strains were grown on PDA, YES, MEA-OX, and CYA agar plates for 10 days, before extraction. Both a “colony extract” and an “agar extract” were made, as described in the Materials and Methods section, and the extracts were analyzed in –ESI mode. AQs were identified by de-replication with the MS/MS spectral library from the GNPS [35].

FBMN was generated from the data and one cluster contained features identified as AQ monomers while another cluster contained features identified as AQ dimers (Figure 6A). Two features with *m*/*z* 537.08 were identified as the [M-H]^−^ adduct of the AQ dimers skyrin (**7**) and isoskyrin (**8**) by the GNPS spectral library. However, both features were identified as skyrin by our in-house spectral library. Because skyrin and isoskyrin are structurally very similar, it is not surprising that the same feature could indicate either two isomers when consulting various spectral libraries. Additionally, three of the neighboring nodes represented isomeric compounds to skyrin and isoskyrin (*m*/*z* 537.08). Two features indicated the gain and loss of a phenol group (*m*/*z* 553.0792 and *m*/*z* 521.0881, respectively) and another indicated the gain of C_2_H_2_O compared to skyrin (*m*/*z* 579.0942). While most of these adjacent features were observed in both *T. islandicus* strains, *m*/*z* 579.0942 was only observed in strain 1. Neither skyrin, isoskyrin, nor any of the connecting nodes were observed in the agar extracts but only in the colony extracts.

In the AQ monomer cluster, three features were identified as the [M-H]^−^ adducts of endocrocin (*m*/*z* 313.0362, **9**), emodic acid (*m*/*z* 299.0205, **10**), and citreorosein (*m*/*z* 285.041, **11**) by the GNPS library. The identity of the endocrocin feature and the citreorosein feature was supported by the in-house library. A connecting node (*m*/*z* 301.0361) between citreorosein and emodic acid suggests a structure with an extra OH group compared to citreorosein and one double bond less than emodic acid. This feature connected to another with a suggested loss of one ketone group (*m*/*z* 287.0567). A connecting node (*m*/*z* 269.0465) to endocrocin indicated the loss of the carboxylic acid group, being structurally identical to emodin (**12**). This feature was indeed identified as emodin by the in-house library. Endocrocin and citreorosein were observed both in the colony extracts and the agar extracts, indicating the secretion of those quinones; however, the distribution of the features in the agar extracts compared to the colony extract was very low (Figure 6C). It should also be mentioned that the colony extracts also contained a portion of the agar and the analysis is thus not a strict comparison between the intra- and extracellular metabolites.

The library scores of the identified AQs are shown in Table 2.

We also investigated the distribution of the features on the various growth media. Most features were observed on all four media types with few exceptions, i.e., the AQ dimer *m*/*z* 579.0942 was only observed on PDA (Figure 7A,B).

Next, we investigated the raw data of the colony extract of *T. islandicus* IBT 11168 grown on YES, as this was the extract with the largest proportion of isoskyrin and endocrocin. AQ dimers with masses of isoskyrin (*m*/*z* 537.0827 and *m*/*z* 553.0792) had several isomeric peaks throughout the –ESI chromatogram (Figure 7C, red and cyan, respectively). The largest of these isomeric peaks constituted some of the largest peaks in the chromatogram along with endocrocin (blue).

Thus, the production of AQs from *T. islandicus* is promising, as these AQs constitute some of the largest peaks in the chromatogram. However, the AQs were primarily intracellular, meaning that mycelium extraction would be required to isolate these compounds.

## 4. Discussion

In this work, we evaluated the fungal production of three types of quinones: TQ, TA, and AQ after an initial screening step. Below, each of the cases are discussed and potential steps for an improvement in the production are suggested. In all cases, screening with a higher number of production strains and testing more culture conditions should always be considered as a first approach for optimization of a specific metabolite. Statistical optimization methods, for example, such as full factorial design, have previously been used to optimize fungal quinone production once the initial screening has been completed [11,41].

Neither TQ nor its hydroquinone form toluquinol were observed in the *P.* cf. *neogriseofulvum* strains investigated, and further experiments on TQ were abandoned. The three strains were selected due to their ability to produce patulin, hypothesizing that they might also produce TQ and toluquinol, which are early shunt products of the patulin biosynthetic pathway. The observation that many other metabolites, including patulin, were produced by the strains suggests that the lack of TQ was not due to compromised fungal growth.

TQ is also part of other biosynthetic pathways, e.g., the yanuthone pathway [8]. Thus, TQ-producing strains could potentially also be found among strains able to produce yanuthones and patulin. If the production of TQ should be pursued further, screening a broader sample of patulin and yanuthone producers would be a logical next step.

Because TQ is a shunt product from the patulin pathway, the two metabolites compete for the same precursors. Thus, if a large amount of patulin is produced, less precursor is available for the production of TQ, all things being equal. Thus, if the production of TQ, should be enhanced, a beneficial step could be to discover or engineer a strain with a partially deleted patulin pathway, ensuring that more of the shunt products could be produced. This would also minimize the potential health risks for working with the production, as patulin is a known mycotoxin [39]. Many quinones, including TQ, have shown very poor ionization by ESI. Thus, it is likely that TQ could be detected if a derivatization strategy was used, as has been shown previously in the literature [42].

For the production of TA, two *Aspergillus* strains were grown on solid medium and as they were both able to produce TA, and they were subsequently tested on liquid medium as surface cultures. We showed that *A. parvulus* was able to secrete TA into the liquid medium and that TA was one of the major metabolites in the chromatogram. It is an interesting observation that *A. christenseniae* was unable to produce TA on liquid medium when it was able to on solid agar medium. This clearly shows that screening on agar medium does not always correlate with the liquid medium. Both strains produced a thick mycelium mat on top of the liquid medium, so the lack of TA in *A. christenseniae* was not likely to be caused by compromised mycelium growth. However, the amount of other metabolites observed in the chromatograms of the *A. christenseniae* strain was generally low compared to what was observed by the *A. parvulus* strain (Figure 5), indicating that the two strains were not equally productive on liquid medium. Repeating the assay with alternative media might have yielded other results and could be an interesting experiment for future studies. The fact that TA was secreted by *A. parvulus* is a valuable feature, as secretion of a target metabolite can be an advantage in a downstream processing set-up, as mycelium extraction is not needed. TA was one of the major peaks in both the –ESI and DAD chromatograms of the supernatant extract of *A. parvulus*, suggesting that this strain is a potential good production host of quinone and further optimization of TA production can be encouraged. Since large amounts of the putative terremutin were also found in the liquid medium, a future strategy could be to engineer the pathway towards TA by overexpressing the last oxidative step going from terremutin to TA.

The last example was the production of AQs. Two *T. islandicus* strains were selected for analysis, as the species is a known producer of both AQ dimers and monomers [6]. We used FBMN to analyze the many AQ analogs produced, including both dimers and monomers. In our studies, we found no extracellular AQ dimers, but the AQ monomers, citreorosein and endocrocin, were observed in low amounts in the agar extracts. In some of the samples, AQs constituted some of the major chromatogram peaks. The *T. islandicus* strains might be good candidates for further optimization, although mycelial extraction of metabolites might be an issue at a large scale, as organic solvent extraction is expensive and not particularly environmentally friendly, and physical treatments such as microwave-assisted and ultrasonication-assisted extraction are either very costly or poorly scalable [43].

Conclusively, while AQ production from *T. islandicus* appears very feasible, the required mycelium extraction step for harvesting quinones means the species has a disadvantage compared to fungi that secrete quinones, as was the case with *A. parvulus.* It might be possible to find growth conditions from which the proportion of secreted citreorosein and endocrocin is enhanced.

## Figures and Tables

**Figure 1 biomolecules-12-01041-f001:**
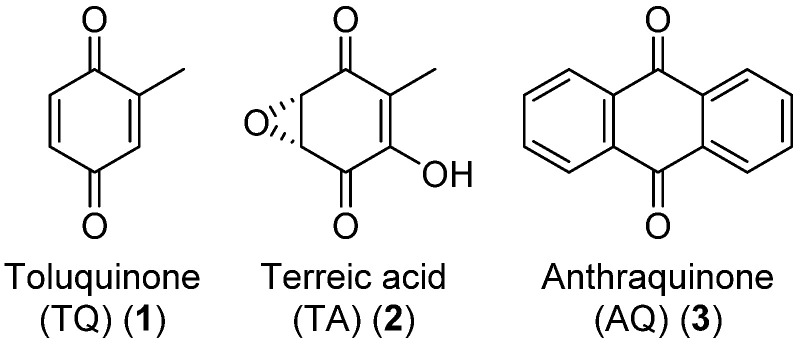
Structures of the target quinones of the presented work. Anthraquinone (AQ) is a large class of quinones with many modifications and here, the simplest structure is presented.

**Figure 2 biomolecules-12-01041-f002:**
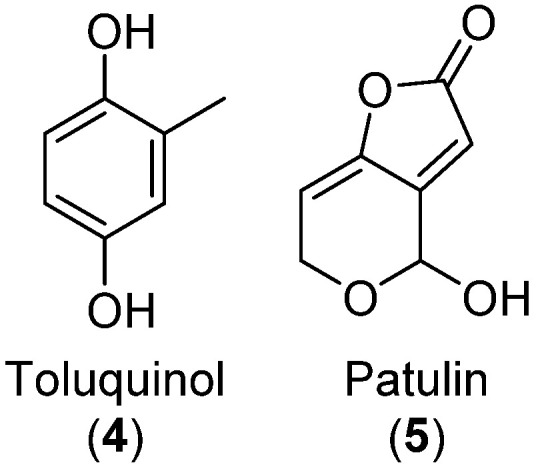
Molecular structure of toluquinol and patulin.

**Figure 3 biomolecules-12-01041-f003:**
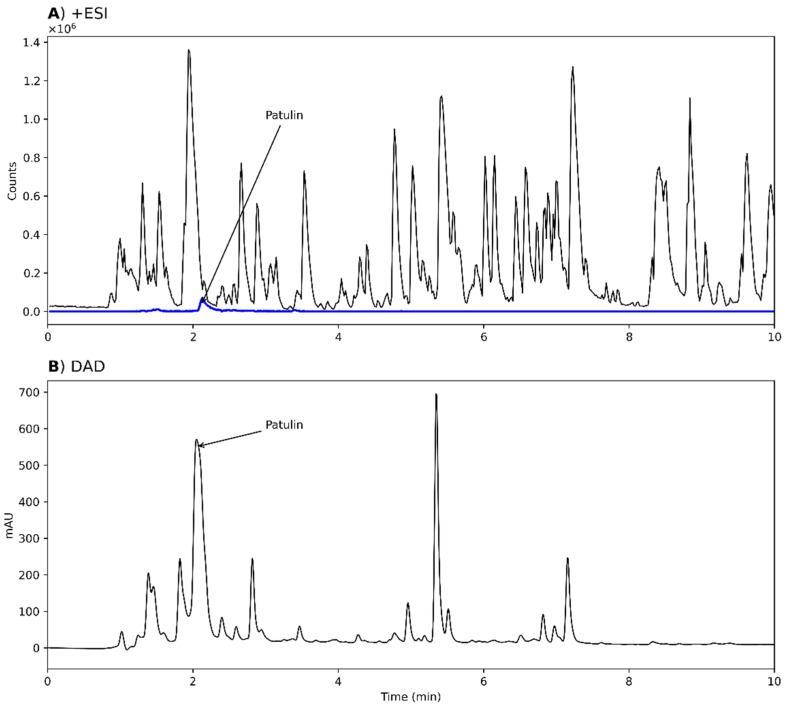
Chromatograms from analysis of an extract of *Penicillium* cf. *griseofulvum* IBT 17755, grown on potato dextrose agar (PDA). (**A**) MS chromatogram from positive electrospray ionization mode (+ESI). An extracted ion chromatogram (EIC) of patulin (**5**, blue, *m*/*z* 155.0339 ± 10 ppm) is overlaid. (**B**) UV-VIS chromatogram from diode array detection (DAD). In both chromatograms, the position of patulin is annotated. As the sample reaches the DAD detector first, there is a retention time delay of 0.068 min between the chromatograms.

**Figure 4 biomolecules-12-01041-f004:**
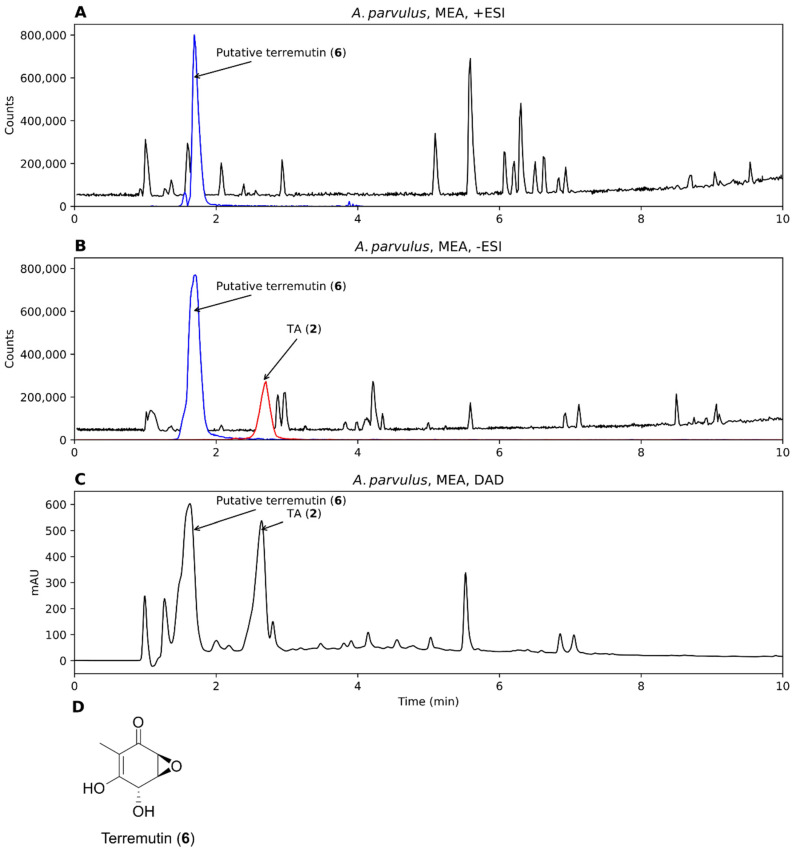
Chromatograms from analysis of an extract of *Aspergillus parvulus*, grown on malt extract agar (MEA), analyzed by +ESI (**A**), -ESI (**B**), and DAD (**C**). EICs of putative terremutin (**6**, *m*/*z* 157.0495 and *m*/*z* 155.0350 ± 10 ppm, representing [M+H]^+^ and [M-H]^−^) and TA (*m*/*z* 153.0193, representing [M-H]^−^) have been overlaid on the chromatograms (blue and red, respectively). (**D**) Molecular structure of terremutin.

**Figure 5 biomolecules-12-01041-f005:**
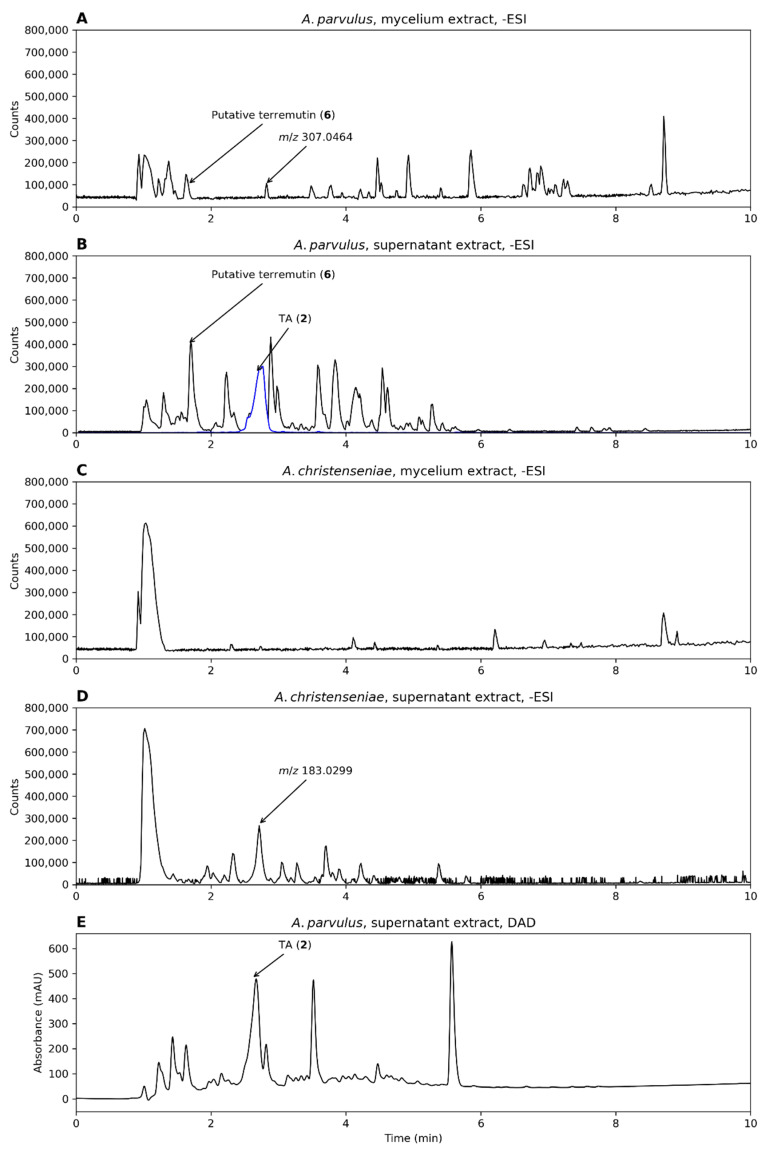
Chromatograms from analysis of *A. parvulus* and *A. christenseniae*, grown on liquid malt extract broth–oxoid brand (MEA-OX). (**A**,**B**) Mycelium and supernatant extracts from *A. parvulus* in –ESI mode and (**C**,**D**) the same for *A. christenseniae*. (**E**) The DAD chromatogram of the supernatant extract of *A. parvulus*. When present, an EIC of the mass of TA (*m*/*z* 153.0193 ± 10 ppm, representing [M-H]^−^) is shown (blue).

**Figure 6 biomolecules-12-01041-f006:**
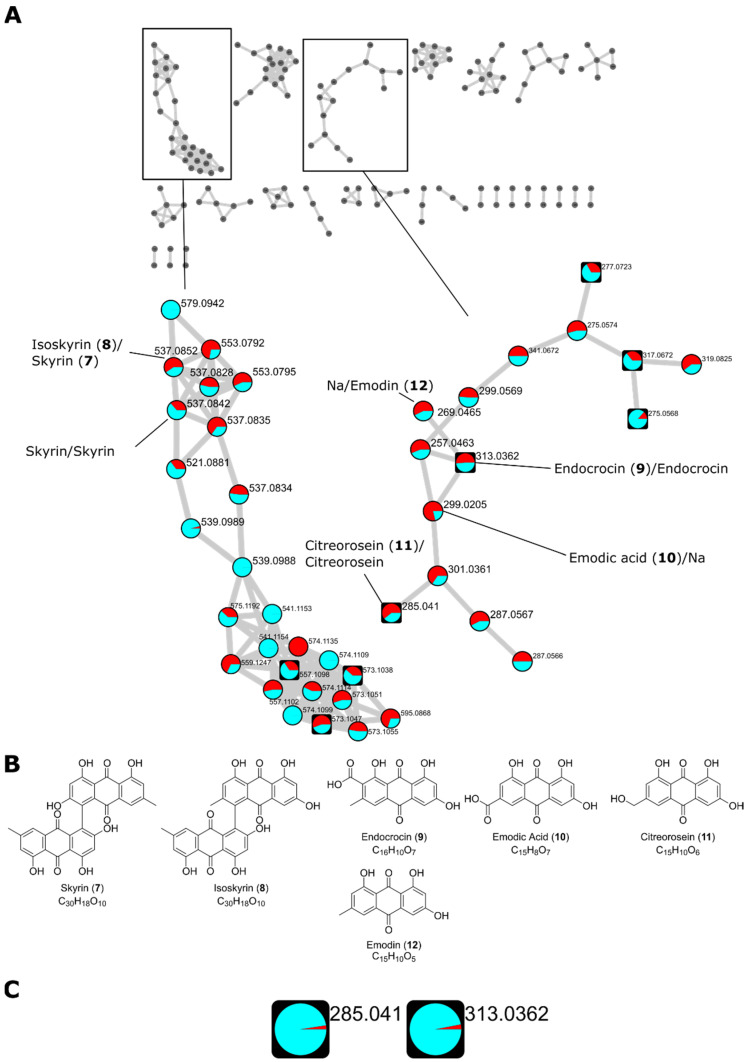
(**A**) Feature-based molecular network (FBMN) of two *Talaromyces islandicus* strains grown on PDA, yeast extract sucrose agar (YES), malt extract agar–oxoid brand (MEA-OX), and Czapek yeast autolysate agar (CYA) agar plates. Two clusters containing library hits for AQ dimers and monomers, respectively, are highlighted. Feature identity is annotated with the results from both databases with the GNPS identification first followed by the in-house library identification separated by “/”. The precursor mass of all identified AQs corresponding to the [M-H]^−^ adduct of the corresponding AQ. A pie chart shows the distribution of the metabolite in *T. islandicus* IBT 20602 and IBT 11168 (cyan and red, respectively). Round nodes represent features only observed in the colony extracts while square nodes represent features that were also observed in the agar extracts. (**B**) Molecular structures of AQs identified by the spectral libraries. (**C**) Distribution of citreorosein (*m*/*z* 285.041 (**11**)) endocrocin (*m*/*z* 313.0362 (**9**)) in the colony extracts (cyan) and agar extracts (red) based on the sum of the precursor ions. Citreorosein and endocrocin showed a distribution of 39.9 to 1 and 31.9 to 1, respectively.

**Figure 7 biomolecules-12-01041-f007:**
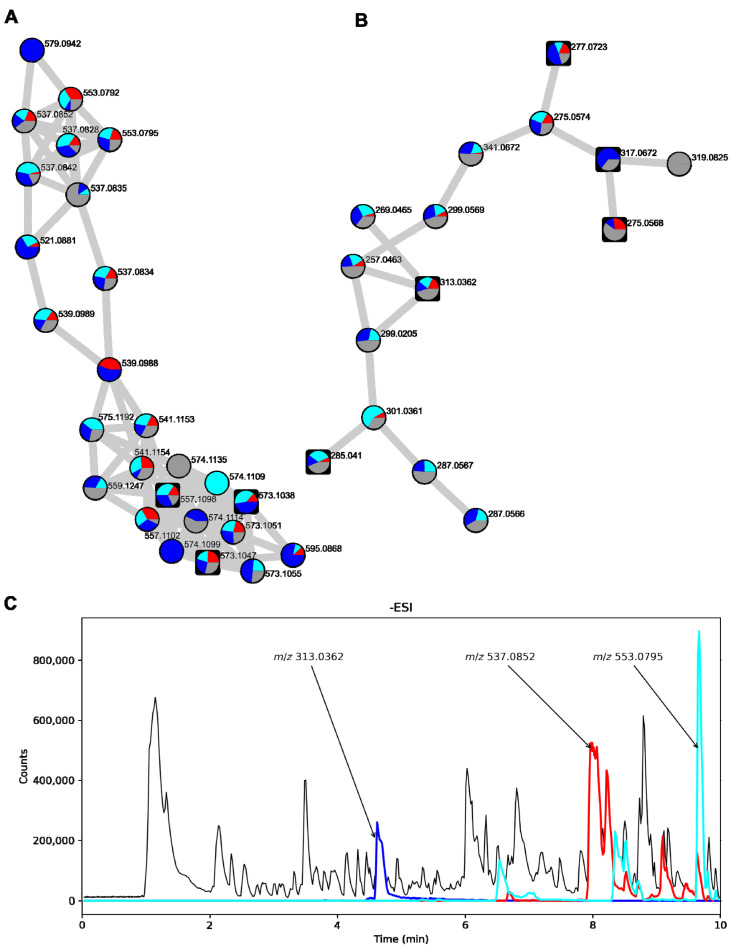
Part of the FBMN of *T. islandicus* strains showing a cluster containing AQ dimers (**A**) and AQ monomers (**B**). Pie charts are labeled on each node showing the distribution of each feature in the different media types used. Red: CYA, cyan: MEA-OX, blue: PDA, grey: YES. Round nodes represent features that were only present in the colony extracts while square nodes represent features that were also observed in the agar extracts. (**C**) Chromatogram of *T. islandicus* IBT 11168 colony extract when grown on YES with –ESI detection. EICs of the accurate masses of the [M-H]^−^ adducts of endocrocin (*m*/*z* 313.0362), skyrin/isoskyrin (*m*/*z* 537.0827), and AQ dimer analog (*m*/*z* 553.0792) ± 10 ppm have been overlaid (blue, red, and cyan, respectively).

**Table 1 biomolecules-12-01041-t001:** Fungal strains used in the presented experiments.

Strain	IBT Number	Target Quinone (s)
*Penicillium* cf. *griseofulvum*	16848	Toluquinone (TQ)
*Penicillium* cf. *griseofulvum*	16849	TQ
*Penicillium* cf. *griseofulvum*	17755	TQ
*Aspergillus parvulus*	22039	Terreic acid (TA)
*Aspergillus christenseniae*	22043	TA
*Talaromyces islandicus*	20602	Anthraquinones (AQs)
*Talaromyces islandicus*	11168	AQs

**Table 2 biomolecules-12-01041-t002:** Identified AQs by the GNPS spectral library and an in-house spectral library.

Precursor Mass (*m*/*z*)	GNPS Identification (Score)	In-House Identification (Score) ^1^
285.041	Citreorosein (0.89)	Citreorosein (99.66)
299.0205	Emodic acid (0.87)	Na
313.0362	Endocrocin (0.77)	Endocrocin (99.66)
537.0842	Skyrin (0.84)	Skyrin (90.8)
537.0852	Isoskyrin (0.83)	Skyrin (99.3)

^1^ Score generated from experimental spectra from *T. islandicus* IBT 11168 grown on YES.

## Data Availability

The FBMN is available from the GNPS website through the following link: https://gnps.ucsd.edu/ProteoSAFe/status.jsp?task=8bcbb547ed734f88813ef84f13c7019c (accessed on 13 July 2022). The mass spectrometry data of *P.* cf. *griseofulvum*, *A. parvulus*, *A. christenseniae* and *T. islandicus* has been deposited at the MassIVE database at the Center for Computational Mass Spectrometry, University of California, and is publically available (website: https://massive.ucsd.edu/ProteoSAFe/static/massive.jsp, MassIVE ID: MSV000089316, accessed on 13 July 2022).

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
