# Peer review of "Production of Fungal Quinones: Problems and Prospects"

_biomolecules, 2022, doi:10.3390/biom12081041_

Round 1

Reviewer 1 Report

This is a very interesting paper on fungal quinones. Authors propose an initial screening of some filamentous fungi to produce three types of important quinones. The manuscript is scientifically sound, and the experiment design is appropriate. The results provide a great advance of the current knowledge of production of the fungal quinones studied. The possibility of producing quinones by fungal cultivation has great prospects since fungi can produce those valuable secondary metabolites on cheap substrates.

Most of the comments refer to the methodology used, which does not always agree with that explained in the results and may need greater concreteness.

In introduction section, authors should make some comments about the fact that some quinones are secreted into the culture media while others are accumulated inside fungal structures. This would aid to understand the different procedures used to detect the studied quinones.

Line 67: Table 1. Write the full name: Penicillium, Aspergillus and Talaromyces. Perhaps it is not necessary to indicate st.1, st 2... since each strain has its IBT collection number. Review the text and figures and replace it with the corresponding IBT number.

Line 76: It is indicated that the plates were incubated 8 days, unless otherwise indicated. But in fact, none of the fungi studied were incubated for 8 days. Thus, to detect TQ and TA the strains were incubated 7 days (lines 176 and 208). In the case of AQs, the strains were incubated for 10 days (line 253).

Line 91: Was a quantification of the spore suspension performed? Which volume of inoculum was added?

Line 194: As five different culture media were assayed you should write “in any of the culture media studied” instead of “in any samples”.

Line 233: It is said for the first time that the strains were inoculated in duplicate. If this procedure was only done on liquid cultures, it should be specified in Material and Methods (line 89).

Line 236: What does it mean “very similar”?

Line 246: MEA-OX (not MEB-OX)

Line 254-255: Does it mean that the agar portion of each plug was separated? Can it be explained better?

Line 318: Please check “Red: CYA, cyan: MEA-OX: cyan, Blue: PDA: blue, Grey: YES”. I think it is not right as cyan and blue are repeated.

Author Response

Thank you for your valuable comments. Our response have been added in bold between each point.

This is a very interesting paper on fungal quinones. Authors propose an initial screening of some filamentous fungi to produce three types of important quinones. The manuscript is scientifically sound, and the experiment design is appropriate. The results provide a great advance of the current knowledge of production of the fungal quinones studied. The possibility of producing quinones by fungal cultivation has great prospects since fungi can produce those valuable secondary metabolites on cheap substrates.

Most of the comments refer to the methodology used, which does not always agree with that explained in the results and may need greater concreteness.

In introduction section, authors should make some comments about the fact that some quinones are secreted into the culture media while others are accumulated inside fungal structures. This would aid to understand the different procedures used to detect the studied quinones.

Such a comment has now been added, with examples and relevant references.

Line 67: Table 1. Write the full name: Penicillium, Aspergillus and Talaromyces. Perhaps it is not necessary to indicate st.1, st 2... since each strain has its IBT collection number. Review the text and figures and replace it with the corresponding IBT number.

Strain numbers have been replaced with IBT numbers, as suggested.

Line 76: It is indicated that the plates were incubated 8 days, unless otherwise indicated. But in fact, none of the fungi studied were incubated for 8 days. Thus, to detect TQ and TA the strains were incubated 7 days (lines 176 and 208). In the case of AQs, the strains were incubated for 10 days (line 253).

The number of incubation days in the methods section has been changed from 8 to 7 now.

Line 91: Was a quantification of the spore suspension performed? Which volume of inoculum was added?

Details about the making of the spore suspension and the inoculation method has been added to the methods section.

Line 194: As five different culture media were assayed you should write “in any of the culture media studied” instead of “in any samples”.

This suggestion has now been incoporated.

Line 233: It is said for the first time that the strains were inoculated in duplicate. If this procedure was only done on liquid cultures, it should be specified in Material and Methods (line 89).

This comment has been implemented.

Line 236: What does it mean “very similar”?

It means that the chromatograms of the extracts were very similar with neglectable differences. This has been clarified in the text.

Line 246: MEA-OX (not MEB-OX)

This has been corrected.

Line 254-255: Does it mean that the agar portion of each plug was separated? Can it be explained better?

The plug methods for the "colony extract" and the "agar extract" have been clarified in the Materials and Methods section, and the specific paragraph referenced in this comment has been rephrased as well.

Line 318: Please check “Red: CYA, cyan: MEA-OX: cyan, Blue: PDA: blue, Grey: YES”. I think it is not right as cyan and blue are repeated.

This has been corrected as well.

Reviewer 2 Report

This work describes the production and identification of quinones in the species Penicillium cf. griseofulvum (3 strains), Aspergillus parvulus, Aspergillus christenseniae and Talaromyces islandicus (2 strains), based on three model quinones: toluqinone (TQ), terreic acid (TA) and anthraquinone (AQ). The study is of interest because the fungal secondary metabolites generally, and the quinone-type compounds in particular, have scarcely been studied in fungi. However, it is important to emphasize that the genera selected for this study have been the matter of recent reviews in the literature.

The methodological description of this study is solid and detailed, which tells of the connection to the analytical methods applied to the extraction and identification of metabolites. However, the experimental design related to the fungal species and the culture conditions used is not clear. The authors divide their study into three parts: the first to study the production of quinone TQ, the second to evaluate the production of TA by species of Aspergillus, and the last to demonstrate the production of AQs. The use of 4 species of filamentous fungi is described (Table 1), and in the case of P. cf. griseofulvum and T. islandicus, 3 and 2 strains of each species were used, respectively. But the justification for the use of these species is not properly established in the manuscript, nor is the use of different strains from the same species justified.

The three models of quinone selected to be studied are not investigated in all the fungal species, and the explanation of this is not in the manuscript. The production of quinones in solid culture medium (in 5 different media) and also in liquid medium is assessed. Nevertheless, it is not clear in the methodology why the production of some quinones (e.g.: TQ) are only evaluated in solid medium and only with one fungal species (P. cf. griseofulvum); by contrast, the production of other quinones (e.g.: TA) was evaluated in solid and liquid medium, but only with Aspergillus. In some cases, 5 different culture media were used, and in other cases only with some. The size of the assays is also not clearly indicated: were all the cultures worked with in duplicate? The metabolite extraction techniques from the solid media were also different, and are not consistent for all the cases analyzed. It is mentioned that the incubation times in the solid medium were 8 days (line 76), but in other places it is indicated that they were 7 days (lines 176, 208) and 10 days (line 253). The different types of fermentation (solid or liquid) and the different culture times could be responsible for great differences between the species/strains of fungi when the secondary metabolites are being produced. In general, the lack of clarity in the design of the experiments affects the understanding of the results. The authors make no mention of the stage of development of the different fungal species in the different culture media (solid and liquid) at the point of extraction from the metabolites. The lack of detection of some of the metabolites studied or low yields could be related to poor mycelial growth, or perhaps to an inadequate incubation period.

In the discussion of the results, the authors indicate that of the metabolites analyzed, neither TQ nor toluquinol were detected in this study, but it would be important to specify that these metabolites were only analyzed in the three strains of P. cf. griseofulvum in solid medium. The authors indicate that A. christenseniae was able to produce TA in solid medium, but not in liquid medium. They also indicate, however, that of the two replicas of this species cultured in liquid medium, only one grew. This assay could have been repeated to confirm the production or lack of production of TA by A. christenseniae based on a duplicate assay. Perhaps it could be a problem associated with the growth of this species in the medium used (only MEA-OX)? For that, a mention of the stage of development reached by the fungi in the different culture media would have been interesting. The authors indicate in their discussion the alternative of engineering the metabolic pathways as strategies to improve the production of quinones by these fungal species. Indeed, that may be an excellent approach. Nevertheless, it is likely that through a simpler approach, adjusting the culture conditions, solid or liquid medium, it would be possible to optimize the production of the quinones studied by the fungal species used here.

Other issues:

-There is repetition of the methodological description in the item Results. Please review.

-The last part of the discussion (lines 360-374) veers away from the central subject of this study, since different methods of metabolite extraction were not evaluated here.

-There are figures that could be presented as complementary material (e.g.: Fig. 2 and 5)

-Review line 318 of the legend of Fig. 8, the nomenclature is not unclear.

Author Response

Thank you for your valuable comments. Our response is added in bold below each point.

This work describes the production and identification of quinones in the species Penicillium cf. griseofulvum (3 strains), Aspergillus parvulus, Aspergillus christenseniae and Talaromyces islandicus (2 strains), based on three model quinones: toluqinone (TQ), terreic acid (TA) and anthraquinone (AQ). The study is of interest because the fungal secondary metabolites generally, and the quinone-type compounds in particular, have scarcely been studied in fungi. However, it is important to emphasize that the genera selected for this study have been the matter of recent reviews in the literature.

The methodological description of this study is solid and detailed, which tells of the connection to the analytical methods applied to the extraction and identification of metabolites. However, the experimental design related to the fungal species and the culture conditions used is not clear. The authors divide their study into three parts: the first to study the production of quinone TQ, the second to evaluate the production of TA by species of Aspergillus, and the last to demonstrate the production of AQs. The use of 4 species of filamentous fungi is described (Table 1), and in the case of P. cf. griseofulvum and T. islandicus, 3 and 2 strains of each species were used, respectively. But the justification for the use of these species is not properly established in the manuscript, nor is the use of different strains from the same species justified.

The reasoning has now been added to the Materials and Methods section, as well as commented on in the Discussion section.

The three models of quinone selected to be studied are not investigated in all the fungal species, and the explanation of this is not in the manuscript.

The model quinones are not expected to be produced by all the fungal species investigated, thus only the specific target of the species were investigated in each case. Table 1 has been expanded to include the target quinones for every strain.

The production of quinones in solid culture medium (in 5 different media) and also in liquid medium is assessed. Nevertheless, it is not clear in the methodology why the production of some quinones (e.g.: TQ) are only evaluated in solid medium and only with one fungal species (P. cf. griseofulvum); by contrast, the production of other quinones (e.g.: TA) was evaluated in solid and liquid medium, but only with Aspergillus.

The P. cf. griseofulvum strains were unable to produce TQ on solid medium and therefore subsequent experiments were not performed. With the Aspergillus species, both produced TA on solid medium and thus liquid medium was the next step in the screening process. This has been clarified in the Discussion section.

Having a broader array of strains could be beneficial for TQ production and a sentence detailing a broader selection of species has been added to the Discussion section, as a potential next step.

The size of the assays is also not clearly indicated: were all the cultures worked with in duplicate?

Only the liquid cultures were cultivated in duplicates. From the agar plates, only one colony of each condition was selected for extraction, as quantification was not a goal of this study. This has now been clarified in the Materials and Methods section.

The metabolite extraction techniques from the solid media were also different, and are not consistent for all the cases analyzed.

When the different methods were used has now been clarified in the Materials and Methods section.

It is mentioned that the incubation times in the solid medium were 8 days (line 76), but in other places it is indicated that they were 7 days (lines 176, 208) and 10 days (line 253).

The 8 days incubation has been changed to 7 days in the Materials and Methods section to avoid confusion. The AQ strains grew for 10 days, yes.

The different types of fermentation (solid or liquid) and the different culture times could be responsible for great differences between the species/strains of fungi when the secondary metabolites are being produced.

Yes, different culture times and conditions affects the metabolite profiles. However, for the strains in the individual cases, fungi were grown and extracted by the same method (e.g. the P. cf. griseofulvum strains were all grown for 7 days).

In the case of the TA producing strains, A. parvulus grew on MEA, MEA-OX and PDA while A. christenseniae only grew on MEA and MEA-OX plates. In reality, A. christenseniae were also cultivated on PDA, but the extraction vial was compromised during the extraction step by accidentally dropping the vial. Thus the extract was omitted from the study. As we acknowledge that this difference can be odd, and due to the fact that the data from the PDA plate is never used, we decided to omit the PDA plate from the study such that both Aspergillus strains were cultured on the same medium in the paper.

In general, the lack of clarity in the design of the experiments affects the understanding of the results. The authors make no mention of the stage of development of the different fungal species in the different culture media (solid and liquid) at the point of extraction from the metabolites. The lack of detection of some of the metabolites studied or low yields could be related to poor mycelial growth, or perhaps to an inadequate incubation period.

When the Aspergillus species grew on liquid medium a thick mycelium mat formed on top of the medium for the replicates that grew, indicating that the lack of TA by P. christenseniae under these conditions were not due to lack of mycelium. This observation has been added to the Results section and the Discussion section. In regards to P. cf. griseofulvum the plethora of other metabolites in the chromatograms does not indicate that the lack of TQ, was due to compromised fungal growth. This observation was also added to the Discussion section.

In the discussion of the results, the authors indicate that of the metabolites analyzed, neither TQ nor toluquinol were detected in this study, but it would be important to specify that these metabolites were only analyzed in the three strains of P. cf. griseofulvum in solid medium. The authors indicate that A. christenseniae was able to produce TA in solid medium, but not in liquid medium. They also indicate, however, that of the two replicas of this species cultured in liquid medium, only one grew. This assay could have been repeated to confirm the production or lack of production of TA by A. christenseniae based on a duplicate assay. Perhaps it could be a problem associated with the growth of this species in the medium used (only MEA-OX)? For that, a mention of the stage of development reached by the fungi in the different culture media would have been interesting.

Both strains produced thick mycelium mats on top of the liquid culture, as mentioned above, suggesting that the lack of TA observed was not due to poor mycelium growth. However, the ESI and DAD chromatograms of A. christenseniae grown on liquid culture, were considerably less metabolite rich than the corresponding chromatograms of A. parvulus. Thus, if another medium than MEA-OX or incubating for more than 11 days, might have given different results. However, the study represents initial screening efforts and the issues that might arise, so a thorough investigation is outside the scope of the article. The issue has been mentioned in the Discussion section.

The authors indicate in their discussion the alternative of engineering the metabolic pathways as strategies to improve the production of quinones by these fungal species. Indeed, that may be an excellent approach. Nevertheless, it is likely that through a simpler approach, adjusting the culture conditions, solid or liquid medium, it would be possible to optimize the production of the quinones studied by the fungal species used here.

Indeed, medium optimization studies is arguably the easiest and first optimization approach that could be taken in all the different cases. This has been clearly stated in the Discussion section now.

Other issues:

-There is repetition of the methodological description in the item Results. Please review.

This has been addressed and several passages have been deleted or re-written.

-The last part of the discussion (lines 360-374) veers away from the central subject of this study, since different methods of metabolite extraction were not evaluated here.

Agreed, that part was a discussion on mycelial extraction and turned out to be way too long and out of the scope of the article. This has now been corrected.

-There are figures that could be presented as complementary material (e.g.: Fig. 2 and 5)

The MS/MS figures have now been moved to Supplementary Materials as suggested.

-Review line 318 of the legend of Fig. 8, the nomenclature is not unclear.

This has been corrected.

Round 2

Reviewer 2 Report

In this new version of the manuscript the authors most of the suggested changes. The methodology was improved.